# Neuroinflammatory Proteins in Huntington’s Disease: Insights into Mechanisms, Diagnosis, and Therapeutic Implications

**DOI:** 10.3390/ijms252111787

**Published:** 2024-11-02

**Authors:** Xinhui Li, Huichun Tong, Shuying Xu, Gongke Zhou, Tianqi Yang, Shurui Yin, Sitong Yang, Xiaojiang Li, Shihua Li

**Affiliations:** 1Guangdong Key Laboratory of Non-Human Primate Research, Key Laboratory of CNS Regeneration (Ministry of Education), Guangdong-Hongkong-Macau Institute of CNS Regeneration, Jinan University, Guangzhou 510632, China; lxh1996@stu2022.jnu.edu.cn (X.L.); tonghuichun@stu2021.jnu.edu.cn (H.T.); xushuying@stu.jnu.edu.cn (S.X.); zgk00@stu2022.jnu.edu.cn (G.Z.); yangtianqi12138@stu2021.jnu.edu.cn (T.Y.); yinshurui20040416@stu2022.jnu.edu.cn (S.Y.); yangsitong@stu2023.jnu.edu.cn (S.Y.); xjli33@jnu.edu.cn (X.L.); 2Department of Neurosurgery, The First Affiliated Hospital of Jinan University, Guangzhou 510630, China

**Keywords:** Huntington’s disease, neuroinflammation, glia, cytokines, neuroproteins

## Abstract

Huntington’s disease (HD) is a hereditary neurodegenerative disorder caused by a CAG tract expansion in the huntingtin gene (*HTT*). HD is characterized by involuntary movements, cognitive decline, and behavioral changes. Pathologically, patients with HD show selective striatal neuronal vulnerability at the early disease stage, although the mutant protein is ubiquitously expressed. Activation of the immune system and glial cell-mediated neuroinflammatory responses are early pathological features and have been found in all neurodegenerative diseases (NDDs), including HD. However, the role of inflammation in HD, as well as its therapeutic significance, has been less extensively studied compared to other NDDs. This review highlights the significantly elevated levels of inflammatory proteins and cellular markers observed in various HD animal models and HD patient tissues, emphasizing the critical roles of microglia, astrocytes, and oligodendrocytes in mediating neuroinflammation in HD. Moreover, it expands on recent discoveries related to the peripheral immune system’s involvement in HD. Although current immunomodulatory treatments and inflammatory biomarkers for adjunctive diagnosis in HD are limited, targeting inflammation in combination with other therapies, along with comprehensive personalized treatment approaches, shows promising therapeutic potential.

## 1. Introduction

Huntington’s disease (HD) is a proteinopathogenic neurodegenerative disorder caused by the abnormal expansion of CAG repeats beyond 36 in exon 1 of the *IT15* gene, which encodes the huntingtin (HTT) protein with an expanded polyglutamine (polyQ) in its N-terminus [1]. A pathological hallmark of HD is the abnormal aggregation of mutant huntingtin (mHTT). Specifically, mHTT encoded by exon 1 accumulates abnormally and forms inclusions containing expanded polyglutamine (polyQ) repeats in the cytoplasm and nucleus of a cell [2]. In simple terms, individuals with 36 to 39 CAG repeats have a variable risk of developing HD phenotypes, while 40 or more repeats inevitably lead to the onset of overt symptoms, including motor dysfunction, cognitive decline, and behavioral changes [3]. The recently proposed Huntington’s Disease Integrated Staging System (HD-ISS) builds upon previous scoring systems by more comprehensively summarizing the biological features, clinical manifestations, and functional assessments of HD across four stages: Stages 0, 1, 2, and 3 [4]. Patients with HD can be evaluated with this system when they are detected to have 40 or more CAG repeats. Stage 0 is characterized by an aberrant CAG amplification but no pathophysiological marker alterations or clinical manifestations. Stage 1 refers to individuals in whom biological markers of neurodegeneration (such as imaging signals abnormalities in MRI) are detectable, but without obvious clinical symptoms [4]. This stage is considered a critical period for early therapeutic intervention. The establishment of these standard disease stages in HD underscores the growing importance of biological markers in the diagnosis and treatment of HD and further suggests the future feasibility of screening for early HD through molecular biomarkers such as neurofilament light chain (NfL) [5]. Patients in Stage 2 typically display observable clinical phenotypes such as declines in cognitive or motor function, which can be measured using tests like the Symbol Digit Modalities Test (SDMT) for cognitive function and the full Total Motor Score (TMS) for motor function. As the disease progresses to Stage 3, there are noticeable functional changes that can be assessed using the Total Functional Capacity (TFC) scale and the Independence Scale. Stage 3 of HD can persist for decades [4]. HD typically manifests around the age of 45, with a normalized CAG age product (CAP) score of approximately 100 [6]. However, the course of HD varies significantly among individuals, largely due to the presence of genetic modifiers. The pre-symptomatic stage of HD is generally believed to last between 10 to 15 years [7]. The duration of each stage in HD, especially the pre-symptomatic stage, is heavily influenced by the number of CAG repeats in the *HTT* gene [8]. Individuals with longer CAG repeat numbers may have a shorter time between genetic diagnosis and clinical symptom onset, potentially leading to an earlier onset of the disease [8,9]. The progression of HD is also affected by the instability of somatic CAG repeat variations [10], which adds complexity to predicting the course of the disease. Research has shown that the amplification of CAG repeats sequences to 115 or higher in a specific number of susceptible cells could trigger motor symptoms in HD [11].

Pathologically, medium spiny neurons (MSNs) in the striatum are the primary cell type affected by HD, though HTT is ubiquitously expressed in both glial cells and neurons throughout the central nervous system (CNS) and the whole body [12,13]. The underlying reason for their selective vulnerability remains unclear. In addition to the highly selective neuronal vulnerability, another prominent pathological feature of HD is glial proliferation and functional alteration. Both postmortem brain tissues from patients with HD and HD animal models have shown widespread activation of pro-inflammatory astrocytes and microglia [14,15,16]. However, it remains uncertain whether this is due to cell-autonomous toxicity caused by mHTT expression in glial cells or non-cell-autonomous toxicity arising from an immune response following neuronal damage induced by mHTT.

To better replicate and study the pathological features of HD, various rodent models have been created and studied [17]. It has been verified that there is similar glial pathology between HD patients and mouse models [18,19]. Early researchers induced HD-like pathological features in rodents by injecting toxic chemicals (such as quinolinic acid, QA, or QUIN) or neurotoxic substances, resulting in neuronal cell death and neuroinflammatory responses [20]. Subsequently, transgenic and knock-in (KI) mouse models were developed, characterized by expression of full-length or truncated fragments of the human *HTT* gene. Notable models discussed in this paper include transgenic R6/2, N171-82Q, YAC128, and BACHD mice, as well as full-length *HTT* KI models such as CAG140, zQ175, HdhQ92, HdhQ111, and HdhQ150 mice, alongside models expressing mHTT specifically in different types of glial cells [21,22,23]. While the role of inflammation in HD has been widely reported and summarized, this review focuses on the glial responses and peripheral immune reactions triggered by mHTT in preclinical experiments or clinical trials, highlighting recent literature advances that detail their significance in the pathological mechanisms, diagnosis, and treatment of HD. 

(1) What are the main findings? 

A pro-inflammatory milieu is demonstrably present in the central and peripheral compartments of animal HD models and human HD subjects, whether alive or post-mortem.In HD, microglia and astrocytes are instrumental in inflammation; whether oligodendrocytes are potentially involved in initiating inflammatory response requires additional study.In HD, prolonged neuroinflammation precipitates neurodegeneration and pathological neuronal death, including necroptosis, pyroptosis, ferroptosis, and autophagic dysfunction.

(2) What is the implication of the main finding?

Interventions targeting glial inflammation are emerging as promising treatments, and the combination of gene therapy and immunotherapy may potentially offer synergistic benefits.Quantifying inflammation to monitor HD progression is hampered by the lack of a unified assessment framework.Species-specific differences and the constraints of animal models obfuscate the translation of pathophysiological and immunological findings from bench to bedside.

## 2. Glia-Derived Inflammatory Markers in HD

In addition to the diverse types and functions of neurons, astrocytes, microglia, and oligodendrocytes (OLs) are integral components of the CNS. These glial cells not only provide structural and metabolic support but also serve as resident immune responders, reacting to neuronal damage within the brain. In the subsequent sections, this review will discuss in detail the analysis of the roles played by inflammation-related markers and proteins associated with different glial cell types in the pathology of HD based on a comprehensive review of relevant literature.

### 2.1. Microglia-Derived Inflammatory Markers in HD

Microglia in the CNS are essentially a subtype of macrophages, originating from erythromyeloid progenitor cells during early embryonic development [24]. Another subtype of macrophages in the CNS is known as border-associated macrophages (BAMs), which are primarily located in the meninges, choroid plexus, and perivascular regions, in contrast to microglia that reside within the parenchyma [25,26]. In the field of neurodegenerative diseases (NDDs), microglia-induced inflammation, along with the related pathways and targets, has been widely reported [27]. In recent years, increasing attention has been paid to the inflammatory roles mediated by BAMs in NDDs, such as Parkinson’s disease (PD) and Alzheimer’s disease (AD) [28,29].

Morphologically, microglia can exist in either a resting or activated state, with the former characterized by an irregular ramified structure and the latter exhibiting an amoeboid shape [30]. Functionally, microglia are classified into pro-inflammatory M1 and anti-inflammatory M2 phenotypes [30]. With the advent of single-cell transcriptomics and spatial transcriptomics technologies, the classification of microglia has become increasingly diversified and personalized, giving rise to new terms such as the disease-associated microglia (DAM), the microglial neurodegenerative phenotype (MGnD), and the cross-disease-associated microglia (CDAM) [31,32]. As primary innate immune defenders, microglia interact with their surrounding environment through its various gene-encoded receptors and sensors, enabling rapid responses to pathological damage, such as recognizing damage-associated molecular patterns (DAMPs), phagocytosing protein aggregates, and clearing apoptotic cells and necrotic cell debris [33]. Microglia are capable of secreting pro-inflammatory factors such as tumor necrosis factor-α (TNF-α), interleukin 1, and interleukin 6 and 18 (IL-1β, IL-6, and IL-18). They can also secrete anti-inflammatory cytokines like interleukin-10 (IL-10) and transforming growth factor-β (TGF-β). To recruit immune cells, microglia can release chemokines such as monocyte chemoattractant protein-1 (MCP-1 or C-C motif chemokine ligand 2, CCL2) and C-C motif chemokine ligand 5 (CCL5) [34,35]. Additionally, microglia derived from pluripotent stem cells of HD patients have been found to produce elevated levels of pro-inflammatory factors such as IL-6 and TNF-α when stimulated with lipopolysaccharide (LPS) and interferon-gamma (IFN-γ). This heightened inflammatory response in the microglia of HD patients suggests a dysregulation of the immune response in the context of the disease that could contribute to the neuroinflammation observed in HD [36]. In addition to producing inflammatory factors, microglia derived from pluripotent stem cells of HD patients have been shown to generate reactive oxygen species (ROS) and exhibit increased apoptosis. Furthermore, it has been observed that the supernatants of differentiated striatal neurons derived from pluripotent stem cells of individuals with HD, when subjected to heat-shock stress, can induce apoptosis in microglia. These findings suggest that the death of microglial cells in the context of HD may be influenced by non-cell-autonomous toxicity, indicating a complex interplay between different cell types in the disease pathology [36]. Indeed, while the study highlights the presence of hyper-inflammatory responsiveness, increased oxidative stress, and apoptosis in microglia derived from HD patients, the exact causal relationship between these factors has yet to be fully understood. Further research is needed to elucidate the intricate mechanisms underlying these processes and how they contribute to the pathogenesis of HD. Investigating the interplay between hyper-inflammatory responses, oxidative stress, and apoptosis in HD microglia could provide valuable insights into the disease mechanisms and potential targets for therapeutic interventions aimed at mitigating these detrimental effects. While apoptosis is a well-known route contributing to cell death in HD, necroptosis, a regulated cell death pathway distinct from apoptosis, has emerged as a significant contributor to neuronal loss in HD [37]. It is crucial to delve deeper into the fundamental processes underlying the neuroinflammatory response and neurotoxicity in HD, particularly considering the distinct nature of pro-inflammatory necroptosis as a pathway of cell death separate from apoptosis. A comprehensive understanding of the roles played by mHTT and necroptosis in the disease progression necessitates an examination of the cell-autonomous and non-cell-autonomous damage they induce.

Many previous studies have demonstrated gliosis in the striatum and cortex, as well as an increase in both pro-inflammatory and anti-inflammatory proteins, using mouse HD models and post-mortem brain tissues from HD patients [38]. These findings suggest that in HD-affected brain regions, there is an increase in both morphological changes, such as amoeboid transformation, and the number of activated microglia, as indicated by Iba1 (+) cells [38]. Researchers have further refined the understanding of cortical pathology in HD patients, focusing on changes in microglia within the midcingulate cortex [39]. Interestingly, in HD patients with predominant emotional symptoms, the number of activated amoeboid shape and Iba1 (+) microglia was found to be reduced compared to controls [39]. However, in all HD patients, including motor, emotional, or mixed symptoms, neither the number of Iba1 (+) cells nor HLA-DP/DQ/DR immunoreactivity, a marker of microglial activation, showed significant changes in this region [39]. Notably, the proportion of microglia reaching an activated state was positively correlated with the load of mHTT protein as shown by 1C2 staining [39], suggesting the causative effect of mHTT on microglia activation.

The contribution of mHTT expression in microglia to HD pathology has been further validated in two separate studies [21,40]. In the BACHD mice, the selective knockout of mutant *HTT* in microglia showed no benefit for motor function or neuropathology [40]. Conversely, retaining mutant *HTT* expression in microglia while knocking out mutant *HTT* in other cell types significantly improved neurological function [40]. In another report, microglia-specific expression of mHTT has been shown to enhance the activity of the transcriptional elements PU.1 and C/EBP, thereby promoting the transcription of inflammatory cytokines TNF-α and IL-6 [21]. Both in vitro and in vivo experiments suggested that this microglia-specific expression of mHTT may drive neurons toward degeneration [21].

### 2.2. Astrocytes-Derived Inflammatory Markers in HD

Astrocytes can be categorized by anatomical location into protoplasmic astrocytes in the gray matter and fibrous astrocytes in the white matter, with each type performing distinct functions. Protoplasmic astrocytes have shorter branches and interact with nearby neurons, providing nutritional support and maintaining synaptic function, while fibrous astrocytes have longer branches and are mainly localized to myelinated axons, suggesting a role in the regulation of myelinated axons [41]. In general, astrocytes provide structural support and control over molecular homeostasis to the regulation of blood flow, synaptogenesis, neurogenesis, and development of the nervous system. In terms of reactive states, astrocytes can be classified into neurotoxic A1 and neuroprotective A2 types. A1 astrocytes are activated under pathological conditions by factors secreted from microglia, including complement component 1q (C1q), TNF-α, and interleukin-1α, and are associated with neurotoxicity [42]. In contrast, A2 astrocytes are considered protective astrocytes, which can provide neuroprotection in the CNS, especially playing a critical role in acute neuronal injuries [43,44,45,46]. Reactive astrocytes have been identified as pivotal players in the inflammatory processes of the CNS, where they express a diverse range of receptors, including pattern recognition receptors (PRRs) and cytokine receptors [47].

When mHTT is expressed in astrocytes, it impairs their normal function, such as by binding to specificity protein 1 (SP1) to downregulate the transcription of glutamate transporters [22], which in turn reduces glutamate uptake, disrupting calcium and potassium ion homeostasis [48,49,50,51] and causing metabolic reprogramming that leads to mitochondrial dysfunction and ROS imbalance [52,53]. In summary, astrocyte expression of mHTT results in excitotoxicity and reduced synaptic plasticity [54,55]. Selective overexpression of mHTT in mouse astrocytes caused HD-like mouse behavior abnormalities with severe motor deficits and early death [22]. In our previous study, we overexpressed heat shock protein 70-binding protein 1 (HspBP1) in astrocytes of a mouse model using stereotaxic injection of adeno-associated virus (AAV) [56]. As a consequence, astrocytes accumulated mHTT [56]. Intriguingly, neurological problems grew worse, although there was no neurodegeneration [56]. These indicated a significant but indirect impact of non-cell-autonomous toxicity in HD. On the other hand, genetically eliminating the expression of full-length mutant *HTT* specifically in astrocytes of the BACHD/GFAP-CreERT2 mouse model revealed a substantial amelioration in both motor function and psychological symptoms, implying that particular subpopulations of reactive astrocytes contribute significantly to the pathogenesis of HD [57]. Indeed, astrocyte reactivity represents an early event in HD, with astrocyte activation evident in the striatum of pre-symptomatic HD and escalating as the disease progresses [58]. This activation aligns with the transcriptomic profiling of astrocytes derived from HD patients’ pluripotent stem cells, which revealed a pattern of gene expression that correlates with polyQ length, indicating that the extent of the mutation governs the astrocytes’ behavior [59].

When stimulated by LPS, astrocytes from R6/2 mice secreted higher levels of TNF-α compared to wild-type astrocytes [60]. However, astrocytes in R6/2 mice showed reduced release of CCL5, a crucial chemokine for the development and activity of cortical neurons, which exacerbates HD neuropathology [61]. In HD, multiple signaling pathways and transcription factors, including signal transducer and activator of transcription 3 (STAT3) and nuclear factor kappa-B (NF-κB), orchestrate the activation of astrocytes [62,63]. Elevated NF-κB-p65 activity, a key transcriptional regulator of neuroinflammation, was detected in astrocytes from both HD patients and R6/2 mice, with primary astrocytes from R6/2 mice showing increased IκB kinase (IKK) activity, resulting in prolonged NF-κB activation and heightened inflammatory factor production [62]. Moreover, the Janus kinase 2 (JAK2)-STAT3 pathway, which governs astrocyte reactivity, was activated in the striatal putamen of HD patients [63]. Selectively triggering this pathway in astrocytes using viral gene transfer in CAG140 KI mice reduced mHTT aggregates, neuronal defects, and striatal atrophy, while also enhancing proteolytic activity via lysosomal and ubiquitin–proteasome pathways, as revealed by transcriptomic analysis [63].

### 2.3. Oligodendrocyte-Derived Inflammatory Markers in HD

OLs primarily function to maintain neuronal integrity by forming myelin sheaths around axons, which are essential for efficient electrical signal transduction. Although OLs and their precursors, oligodendrocyte precursor cells (OPCs), are not the brain’s primary immune cells, it is noteworthy that OLs can indirectly trigger inflammation by producing myelin debris when injured, which activate microglia or astrocytes [64]. Under pathological conditions, the OL lineage cells engage with the immune system, with OL proliferation [65], differentiation [66], maturation [67], and apoptosis being regulated by immune signals [68]. On the other hand, IFN-γ-treated OLs can secrete chemokines that influence the function of T cells, dendritic cells, and NK cells [69,70]. NG2(+) OPCs also maintain neuronal and microglial homeostasis by secreting TGF-β2 and hepatocyte growth factor (HGF) [71,72]. Furthermore, OL lineage cells express toll-like receptor 3 (TLR3), which regulates microglial differentiation and phenotype [73]. They also express major histocompatibility complex (MHC) class I and II molecules to present antigens and activate T cells [74,75]. Additionally, they utilize low-density lipoprotein receptor-related protein 1 (LRP1) to phagocytose myelin debris [76], and tumor necrosis factor receptor 2 (TNFR2) to enhance peripheral immune cell infiltration and microglial activation [77,78]. While substantial evidence suggests that OLs interact with the CNS immune system, much of this comes from studies on demyelinating diseases like multiple sclerosis (MS), where immune system dysfunction causes myelin damage [79]. Although some evidence suggests that OLs participate in the neuroinflammation process, OLs-mediated neuroinflammation in other NDDs remains to be further explored.

The long-standing assumption has been that white matter abnormalities in HD are merely an accompanying manifestation of progressive striatal atrophy and neuronal loss. White matter and myelin abnormalities are early pathological features of HD and have been repeatedly shown to precede any phenotypic abnormalities or neuronal loss [80]. The CAG140 KI mouse model, which ubiquitously expresses mHTT protein in various types of brain cells but lacks significant neurodegenerative phenotype, exhibited myelin damage as early as 3 months of age [81]. This included a reduction in myelin protein markers, such as myelin oligodendrocyte glycoprotein (MOG), myelin-associated glycoprotein (MAG), 2′,3′-cyclic nucleotide 3′-phosphodiesterase (CNP), and myelin basic protein (MBP), in the cortex and striatum [81]. The myelin sheaths in the subcortical white matter and striatum of CAG 140KI mice also showed reduced thickness as observed by electron microscopy [81]. Furthermore, studies using Cre/LoXP technology to specifically knock out mutant *HTT* in NG2(+) OPC cell lines in the BACHD HD mouse model have shown that this can rescue myelin thinning [82]. Additionally, single-nucleus RNA sequencing (snRNA-seq) of the cortex and striatum of R6/2 mice and human post-mortem HD tissues has revealed that OPCs were arrested at an intermediate maturation state in HD [83]. This disruption in normal OPC development may adversely affect myelination, potentially compromising neuronal function and connectivity. Therefore, white matter alterations have recently been recognized as an important pathological feature associated with HD, though their etiological role in disease onset and progression remains unclear. To investigate the role of OLs in the pathological manifestations of HD, our team conducted research by developing a transgenic mouse model (PLP-150Q) with OLs-specific expression of mutant *HTT* under the control of the PLP promoter [23]. In these mice, we observed transcriptional suppression of myelin proteins following the impact of mHTT on the transcriptional activity of myelin regulatory factor (MYRF) [23]. The subsequent study found that at 5 months of age, when the PLP-150Q mice exhibited pronounced phenotypes, the level of phosphorylated NF-κB was significantly elevated compared to age-matched control PLP-23Q transgenic mice [84]. Phosphorylated NF-κB regulates immune and inflammatory responses by promoting its translocation from the cytoplasm to the nucleus, where it binds to specific DNA sequences and initiates the transcription of target genes [85]. Data from conditional NF-κB expression specifically in mouse mature OLs and snRNA-seq of primary mouse OLs suggested that NF-κB in OLs was a key mediator influencing myelin protein translation and white matter injury [86]. It is worth investigating whether the expression of mHTT within OLs may lead to the release of DAMPs, which could trigger inflammatory responses. Furthermore, the observed elevation of inflammatory factors such as NF-κB in the brains of mice expressing mHTT universally across all neuronal cells cannot rule out the contribution of OL lineage cells.

## 3. Peripheral Immune Responses in HD

Although HD is primarily a CNS disorder, increasing research indicates that peripheral organs also experience various forms of damage during the progression of HD [87]. This includes cardiac failure, testicular damage, and pancreatic dysfunction, which may be related to the widespread expression of HTT and its mutant forms in peripheral organs of mammals [87]. The activation of peripheral immune responses in HD has been confirmed by numerous studies [88,89,90]. In BACHD mice, elevated levels of inflammatory factors have been detected in various peripheral organs, including the kidneys, heart, liver, and spleen [88]. Previous studies have also demonstrated the activation of the peripheral immune system in HD mouse models, including R6/2, HdhQ150, YAC128, and zQ175 mice [89,90].

In pre-manifest HD gene carriers, the concentrations of peripheral pro-inflammatory chemokines and other cytokines were elevated, potentially due to increased cytokine release from macrophages and monocytes. Early research has shown that compared to healthy controls, cytokine release induced by LPS stimulation in macrophages and monocytes from HD gene carriers is enhanced, and this finding has been replicated in HD mouse models [89,91]. Additionally, it has been demonstrated that the increased levels of peripheral inflammatory factors are associated with the activation of microglia in the CNS of pre-manifest HD gene carriers, suggesting that peripheral immune activation can influence the activation of immune cells in the CNS [92].

The interaction between peripheral inflammation and CNS inflammation affects both physiological and pathological states [93]. Primary inflammatory cytokines originating from the periphery can directly enter the CNS or affect it by activating endothelial cells and perivascular macrophages, which synthesize immune activators that can cross the blood–brain barrier [93]. Moreover, the activation of inflammatory pathways induced by these cytokines can lead to blood–brain barrier damage, a common pathological feature of HD [94]. Under normal conditions, T cells are rarely present in the brain due to the restrictive nature of the blood–brain barrier. However, under disease or inflammatory conditions, T cells can cross the blood–brain barrier and participate in immune responses within the CNS. Direct infiltration of peripheral immune cells into the brain is not a primary feature of HD, but the inflammatory response of the peripheral immune system is indeed increased in HD patients and may be associated with disease progression. In HD KI pig models, interleukin-17A (IL17A) produced by Th17 cells in the brain mediated the significant activation of pathways involving interleukin 17 receptor A (IL17RA), interleukin 17 receptor C (IL17RC), TNF receptor-associated factor 3 interacting protein 2 (TRAF3IP2), an inhibitor of NF-κB kinase subunit gamma and epsilon (IKBKG, IKBKE), which in turn activate microglia and astrocytes in the brain, causing significant synaptic damage [95].

Recently, the role of gut microbiota has also raised interest in HD studies. A recent study examined the gut microbiome in HD gene carriers and found significant differences in microbial diversity compared to healthy controls, and certain gut bacteria were linked to cognitive performance and clinical outcomes in HD carriers, suggesting a potential role of gut microbiota in disease progression [96]. One could assume that changes in gut microbiota may lead to various degrees of peripheral immune system alterations, which can subsequently influence CNS pathology. Evidence from HD patients showed that the genus *Intestinimonas* was linked to clinical measures like functional capacity and IL-4 levels, while the genus *Bilophila* was negatively correlated with the pro-inflammatory cytokine IL-6 [97]. Furthermore, studies have shown that gut dysbiosis can result in immune activation dysregulation, gut barrier dysfunction, and chronic inflammation, leading to various neurodegenerative mechanisms [98]. In the case of gut dysbiosis, key immune changes include functional modulation of dendritic cells and glial cells in the CNS, as well as functional modulation of host immune cells such as effector T cells and B cells in the gut, peripheral blood, and peripheral nervous system [99]. Furthermore, gut microbiota and their metabolites interact with different cell components of the CNS by activating immune signaling pathways [100]. Therefore, chronic neuroinflammation caused by excessive accumulation of lymphocytes, cytokines, and chemokines can lead to the aggregation and accumulation of misfolded proteins within and around neurons [101].

## 4. Biological Markers Associated with Inflammatory Damage in HD

In NDDs, pathological neuronal death mechanisms have been confirmed to include necroptosis, pyroptosis, ferroptosis, and autophagic dysfunction [102,103]. These pathways in HD are closely associated with inflammatory damage.

Necroptosis, pyroptosis, and ferroptosis are directly linked to inflammation, sharing common features such as robust microglial activation, increased levels of inflammatory cytokines, and the expansion of the inflammatory response [103]. During necroptosis, the rupture of the cell membrane leads to the release of intracellular contents and DAMPs, which rapidly activate microglia and macrophages, triggering a strong inflammatory response [104]. Pyroptosis is induced through the NOD-, LRR- and pyrin domain-containing protein 3 (NLRP3)/apoptosis-associated speck-like protein containing a CARD (ASC)/caspase-1(CASP1) inflammasome pathway, resulting in the release of pro-inflammatory cytokines such as IL-1β and IL-18 [105]. During the ferroptosis process, the accumulation of lipid peroxides and dysregulated iron metabolism activates inflammatory signaling pathways and microglia, further promoting inflammation [106]. A study has shown that in HD-N171-82Q mice, arachidonate 5-lipoxygenase (ALOX5) -dependent ferroptosis occurred under ROS [107]. In contrast, the relationship between ROS and autophagic dysfunction with inflammation is more complicated. Mitochondrial dysfunction, although initially related to metabolic dysregulation, triggers oxidative stress, which activates inflammatory pathways through nuclear factor NF-κB, thereby promoting microglial activation and the release of pro-inflammatory cytokines, exacerbating inflammation [108]. While autophagic dysfunction is not inherently linked to inflammation, its impairment leads to the accumulation of abnormal proteins, such as mHTT, which activates microglia and astrocytes, inducing neuroinflammation [109]. In a study using MSNs reprogrammed from fibroblasts of HD patients, researchers identified miR-29b-3p, a microRNA whose expression increases with age, as a key factor promoting neurodegeneration by inhibiting autophagy through its interaction with the STAT3 3′ untranslated region [110].

In the continuous pursuit of enhanced comprehension and treatment of HD, researchers are actively engaged in identifying dependable biomarkers to track disease advancement. Biomarkers linked to inflammation-related harm, whether through direct or indirect pathways, not only carry therapeutic significance but also offer potential utility in forecasting and detecting HD. An example of such a biomarker is kynurenine 3-monooxygenase (KMO), which is associated with mitochondrial function and neurotoxicity [111]. KMO is highly expressed in the mitochondrial membranes of microglia and plays a key role in the kynurenine pathway (KP). This pathway produces neurotoxic metabolites such as 3-hydroxykynurenine (3-HK) and QA, along with the neuroprotective metabolite kynurenic acid (KYNA). These productions collectively contribute to the regulation of neuroinflammation, excitotoxicity, OS, and mitochondrial dysfunction [112]. Kynurenine 3-monooxygenase (KMO) has been recognized as a promising therapeutic target in NDDs. Early investigations have revealed heightened levels of toxic metabolites in the neostriatum and neocortex of HD patients [113], as well as in the striatum and cortex of R6/2, YAC128, HdhQ92, and HdhQ111 HD mouse models [114], underscoring the potential significance of targeting KMO in the management of HD. Comprehending the dynamics of KP metabolites in peripheral biofluids is also essential to uncovering their potential as biomarkers and therapeutic targets in NDDs such as HD [115,116]. While the modification of KP metabolites in peripheral biofluids presents intriguing possibilities for HD diagnosis and treatment targets, this discovery remains a subject of debate. Recent studies have indicated that KP metabolites did not show significant differences in the cerebrospinal fluid (CSF) and blood between HD patients and healthy controls [117]. The discrepancy regarding changes in KP metabolites could be attributed to the intricate in vivo processes involved. This pathway is susceptible to various influences, including disease stage, genetic background, environmental factors, and pharmacological treatments. Additionally, metabolite levels in peripheral blood and CSF are subject to fluctuations over time, and samples may have been collected at varying time points across different studies, potentially contributing to inconsistencies in findings.

YKL-40, encoded by the Chitinase 3-like 1 (CHI3L1) gene in humans, is a glycoprotein predominantly derived from astrocytes and is elevated in various NDDs [118]. It plays a pivotal role in neuroinflammatory processes, cellular remodeling, and tissue repair. In the murine model, CHI3L1 (also known as breast regression protein-39, BRP-39), the homolog of human YKL-40, shows high structural and functional similarity [119,120]. BRP-39 has been found to be significantly elevated in the plasma and CSF of pre-symptomatic R6/2, YAC128, and zQ175 mice [121]. Elevated levels of YKL-40 have been detected in the CSF of both manifest HD and pre-symptomatic HD patients, with a clear upward trend compared to control groups, although its concentration in blood has not yet been thoroughly studied [122,123]. Currently, YKL-40 is regarded as a promising potential biomarker for early detection of HD.

Under the influence of inflammatory mediators produced by glial cells, neurons can release intracellular structural components due to functional and structural damage (Figure 1). Among these, NfL and tau proteins, detectable in blood or CSF, as well as phosphodiesterase 10A (PDE10A) measured by positron emission tomography (PET), have been shown to be useful in diagnosing and predicting HD clinical prognosis [124,125]. Neurofilaments and tau are components of the neuronal cytoskeleton. NfL is a major component of the intermediate filaments in myelinated axons of the central and peripheral nervous systems, crucial for maintaining neuronal structural integrity, while tau protein is a microtubule-associated protein that is important for stabilizing microtubules within neurons [126,127]. Phosphodiesterases (PDEs) are key regulators of cyclic nucleotide signaling, with PDE10A being highly expressed in the striatum and capable of degrading cyclic adenosine monophosphate (cAMP) and cyclic guanosine monophosphate (cGMP), thereby regulating neuronal signal transduction [128]. All three proteins are associated with NDDs, including HD. NfL levels in CSF and plasma are similar, and their levels are positively correlated with changes in brain volume and clinical manifestations in HD patients [129]. Elevated NfL levels indicate its potential as a biomarker for neuroinflammation and neuronal damage. For instance, higher NfL levels observed in juvenile HD patients suggest that NfL may be a useful early diagnostic tool and prognostic indicator [130,131]. In a different study, researchers used enzyme-linked immunosorbent assay (ELISA) to demonstrate a correlation between CSF tau protein levels and motor function in HD patients. Significant differences were observed when comparing healthy individuals and HD gene carriers [132]. Hyperphosphorylated tau protein leads to cellular dysfunction and abnormal deposition in the brains of late-stage HD patients, highlighting its importance as a disease progression marker [133], suggesting that therapeutic strategies targeting tau protein may become a potential treatment for HD. In addition to assessing tau levels, visual monitoring through PET has also provided insights into the brains of HD patients. For instance, changes in PDE10A levels may precede evident brain atrophy, suggesting its potential as a diagnostic and prognostic marker [134,135,136,137], thus offering possibilities for early disease recognition and progress monitoring.

Research on these biomarkers is crucial for understanding the pathological processes of HD and offers potential targets for early detection and new treatment development. However, it is important to note that despite the promising potential of these biomarkers, further research is needed to validate their accuracy and reliability before they can be widely applied in clinical settings.

## 5. HD Treatment Drugs Targeting Inflammatory Proteins

Given that neuroimmune inflammation is a key feature of HD and is often detectable in preclinical stages [138], targeted therapy early in the disease may be feasible (Table 1 and Figure 2). Several research hotspots exist for drugs targeting glial cell-related inflammation or immune modulators, including C1q-antibody, semaphorin receptors, and laquinimod (LAQ) (Table 1).

C1q can activate the classical complement cascade and abnormal accumulation of C1q drives complement activation, leading to chronic neuroinflammation and neuronal injury in HD mice [162,163]. In the zQ175 HD mouse model, targeting C1q with ANX-M1 or specifically knocking out the complement receptor C3R on microglia has been shown to reverse synaptic loss and excitatory signal input defects in the striatum [139]. ANX005, a humanized monoclonal antibody targeting C1q, has demonstrated good tolerance and safety in a Phase 2 clinical trial (NCT04514367) [140]. C1q activity and the classical complement pathway were completely blocked during the 6-month treatment, indicating the targeting effect of ANX005 [140]. Furthermore, a swift and continuous reduction in neuroinflammation was observed, indicated by decreased CSF C3a, C3, and YKL-40 levels in HD patients compared with controls [140]. Although secondary outcomes like the Composite Unified Huntington’s Disease Rating Scale (cUHDRS) and TFC showed trends of clinical improvement; however, this treatment did not reach statistical significance [136]. Currently, C1q antibody (ANX005) is about to enter Phase 3 clinical trials by Annexon Biosciences [164].

Semaphorin 4D (SEMA4D) has significant influences on astrocytes. Activation of SEMA4D leads to glial cell activation, inhibits glial cell migration and replacement, and disrupts blood–brain barrier integrity [165]. Pepinemab (VX15/2503), a human IgG4 monoclonal antibody that blocks SEMA4D, has shown some promising results. Antibodies against SEMA4D improved striatal and cortical atrophy and behavioral symptoms in YAC128 mice but did not alleviate motor symptoms [141]. In a Phase 2 trial (SIGNAL, NTC02481674), pepinemab was well tolerated, with 26% relief in caudate nucleus atrophy and significant metabolic level increase in early symptom groups, although improvements in several motor tests did not achieve statistical significance [142].

Oral immunomodulator LAQ can activate the binding site of L-kynurenine (Kyn) in the KP, i.e., the aryl hydrocarbon receptor (AHR), thereby regulating the immune response in the CNS [166]. LAQ could inhibit pro-inflammatory microglia and astrocyte activation in the CNS, particularly in MS, by upregulating microRNA-124a and downregulating the NF-κB pathway, respectively [167,168]. Also, LAQ could diminish inflammation generated by peripheral monocytes in MS [169]. In contrast, in the field of HD, there is currently a lack of evidence for LAQ’s ability to reduce CNS inflammation. Nevertheless, in vitro experiments have confirmed that LAQ decreases the release of hyper-responsive cytokines from peripheral blood monocytes of symptomatic and pre-symptomatic HD patients under LPS stimulation [143]. Interestingly, their results also suggested that the effect may be independent of the downstream pathways of NF-κB [143]. In in vivo experiments, LAQ has shown distinct effects in mouse HD models and human HD patients. In rodent HD models, LAQ was beneficial in improving motor function in R6/2 and YAC128 HD mice [144,145], and myelin recovery and protective effects in PLP-150Q and YAC128 HD mice [84,146]. On the other hand, in HD patients, the LEGATO-HD Phase 2 trial (NCT02215616) by TEVA demonstrated good safety and tolerance of LAQ and effectively reduced caudate nucleus atrophy [147]. However, this intervention did not lead to an improvement in the total motor scores on the Unified Huntington’s Disease Rating Scale–Total Motor Score (UHDRS-TMS) [147]. Meanwhile, LAQ did not result in a decrease in microglial activation in HD patients, as assessed by the microglia-associated neuroinflammation marker translocator protein (TSPO) using PET imaging [148]. Considering the findings derived from both in vitro and in vivo investigations, it is evident that the importance and efficacy of LAQ in alleviating CNS central and peripheral inflammation in HD still need to be further determined.

Another earlier hot topic in HD treatment research is minocycline, an antibiotic drug, distinguished as the sole immunomodulatory agent to undergo complete Phase 3 trials (NCT00029874 (Phase 1/2), NCT00277355 (Phase 2/3)) [151,152]. It has been shown to inhibit the release of inflammatory mediators by microglia in NDDs, such as reducing pro-inflammatory cytokines (e.g., TNF-α, IL-1β, IL-6) and inducing iNOS, thereby mitigating neuroinflammation [170]. It also inhibits caspase activity, blocking some cell death signaling pathways, and reducing neuronal apoptosis or death [171]. In animal experiments, minocycline has been controversial; some researchers argue its ineffectiveness in reducing mHTT burden in R6/2 models at therapeutic doses [149], yet it demonstrates good anti-inflammatory, antioxidant, and neuroprotective abilities in QA-induced HD rodent models [150]. These results suggest that small rodent animal models react to the same treatment differently, which may result from different physiological and metabolic conditions. In human trials, minocycline showed good safety and tolerability in Phase 2 clinical trials [151]. Unfortunately, the Phase 3 clinical trial failed to improve functional outcomes in HD patients [152].

Among various cytokines, IL-1β, IL-6, and TNF-α are notably elevated in affected brain regions and peripheral biofluids in HD [138]. These inflammatory proteins are primary targets for HD treatment and are tested in animal models. The NLRP3 inhibitor MCC950 effectively reduces downstream IL-1β and IL-18 production by inhibiting the NLRP3 inflammasome pathway in microglia, improving motor dysfunction and neuronal survival in R6/2 mice and reducing inflammatory glial cells [153]. IL-6, an inflammatory cytokine that triggers the acute phase response, exacerbated HD-related behavioral phenotypes when the IL-6 gene was knocked out in R6/2 mice and snRNA-seq reveals dysregulation of genes related to synaptic function and neurotrophic tyrosine kinase receptor type 2 (Ntrk2, also known as the brain-derived neurotrophic factor receptor) [172]. Early studies also showed that injecting lentivirus-overexpressing IL-6 or IL-6/IL-6 receptor (IL-6R) into the striatum of rats alleviated QA-induced striatal damage, suggesting that increased activity of IL-6 may have beneficial effects in HD animal brains [154]. TNF-α, one of the major mediators of the neuroinflammation associated with neurodegeneration, was found to increase excitatory synaptic strength and decrease inhibitory synaptic strength in YAC128 HD mice [173]. Using the TNF-α inhibitor, Etanercept, in R6/2 mice reduced brain atrophy but did not improve related motor functions and cognitive deficits [155]. However, current efficacy data for cytokine antagonists or supplements in treating HD were generated in animal models and lack clinical validation, necessitating further experiments to assess the impact of reducing inflammatory factors on animal models or patients with HD.

Existing clinical trials like tominersen (also known as RG6042), based on antisense oligonucleotides (ASOs) technology, and AMT-130, utilizing RNA interference (RNAi) technology, aim to reduce pathogenic mHTT protein levels by binding to *HTT* mRNA [174]. The results of the Phase 1/2a trial (NCT02519036) conducted by Ionis Pharmaceuticals and F. Hoffmann-La Roche back in 2015 showed that treatment with the highest dose of intrathecal injection of tominersen (120 mg) resulted in the mean CSF level of mHTT in HD patients being reduced by 38% [156]. However, Roche announced the termination of the Phase 3 study of the GENERATION HD1 trial of tominersen (NCT03761849) in 2021 [157]. This decision was made due to significantly worse symptoms and higher levels of NfL observed in the high-dose group (120 mg) compared to the placebo group. The termination followed a pre-planned risk/benefit assessment conducted by an independent data monitoring committee. Following the preliminary results and assessment, the GENERATION HD2 trial (NCT05686551) was initiated in 2023. The objective of this trial is to investigate the use of tominersen in treating young HD patients in the prodromal and early stages of the disease [158]. Recently, uniQure has developed the AMT-130 gene therapy, which utilizes a microRNA (miRNA) designed to target both normal and mutant *HTT* mRNA. This miRNA is delivered using an adeno-associated virus serotype 5 (AAV5) vector through stereotactic injection into the brain’s caudate and chiasma nuclei of the striatum in patients with HD. This treatment has shown well tolerance and promising results in preclinical research by reducing levels of both mutant and wild-type HTT and improving motor function [175,176,177]. The AMT-130 Phase 1/2 trials NCT04120493 and NCT05243017 were conducted in the U.S. and Europe, respectively. Following 24 months of follow-up, uniQure’s report in July 2024 indicated that HD in both trials exhibited reduced mHTT levels, improved motor function, and decreased levels of NfL in the CSF [159]. Notably, the high-dose group receiving 60 trillion vector genomes experienced an 80% delay in disease progression compared to the external control group [159]. Both nucleic acid therapies, tominersen and AMT-130, come with certain drawbacks. Tominersen requires monthly intrathecal injections, which can increase the risk of CNS infection and inflammatory responses. On the other hand, while AMT-130 only necessitates a one-time surgical stereotactic injection, there is still a higher likelihood of CNS inflammation due to the invasive nature of the procedure and ongoing viral effects. To address these concerns and minimize adverse effects, the next step of the trial (NCT04120493) will involve a subset of HD patients receiving AMT-130 along with immunosuppressive agents both before and after surgery. This approach is expected to yield valuable outcome data by mid-2025 [159].

Interestingly, some newly tested treatments unexpectedly show effects in reducing brain inflammation. CRISPR/CAS9 technology combined with viral gene delivery has been widely applied in treating NDDs [178]. Our team found that targeting the *HTT* gene with AAV carried guide RNA and Cas9 through intracranial or intravenous injection not only effectively reduced mHTT protein in HD KI pigs and alleviated neuronal damage and functional deficits but also had some reversing effects on inflammation indicators in the striatum, which showed lowering elevated *IL1B* and *NFKB* gene levels in HD KI pigs’ brains [160]. The emerging intrabody technology is using a smaller intracellular-specific antibody to target mutant protein, with many advantages compared to monoclonal antibodies [179]. Our team has discovered that brain and intravenous delivery of AAV-packaged intrabody SM3, an intrabody-derived small fragment, can effectively target intracellular mHTT for lysosomal degradation [161]. This approach has demonstrated a significant reduction in mHTT protein in both in vivo and in vitro, along with notable decreases in Iba1 (+) and GFAP (+) cells in the striatum of CAG140 KI mice [161].

## 6. Future Perspectives

In this review, we have summarized the glial cell and peripheral immune responses triggered by mHTT in HD and listed some neuronal damage biomarkers detectable following inflammatory stimuli. The primary challenge now is whether interventions targeting inflammation can effectively modify the progression and prognosis of HD, potentially delaying the onset of clinical symptoms and extending survival. Despite the notable success of immune modulators in other NDDs, anti-inflammatory or immunomodulatory therapies for HD have yielded limited results in clinical trials. However, this does not preclude the significant potential of immunomodulatory agents in treating HD. Several key questions remain to be addressed: (1) Synergistic effects of combined therapies; can combining immune or inflammation-related treatments with existing gene-targeting technologies offer greater benefits? (2) Targeting specific glial population; could focusing on specific types of glial cells—such as disease-associated microglia, astrocytes, or previously underexplored OLs—emerge as novel therapeutic targets? (3) Translating findings associated with cytokines to humans; is there therapeutic potential in translating cytokine targets that have shown efficacy in HD mouse models to human patients?

Few studies have given some helpful answers to the above questions in mouse models [180,181]. For example, in R6/2 HD mice, researchers using pexidartinib to deplete microglia showed improvements in motor function, reduced mHTT accumulation, and prevention of astrogliosis and striatal volume loss [180]. The study also revealed that microglia play a key role in regulating perineuronal nets, further emphasizing their importance in HD progression and the maintenance of neural structure [180]. However, microglia play vital roles in the CNS and long-term or complete depletion of these cells can lead to detrimental consequences. Furthermore, the inflammation-related cytokines enveloped by extracellular vesicles secreted by microglia, astrocytes, and OLs in NDDs (including HD) have dual roles: they can either protect against or promote disease progression by spreading toxic signals and contributing to neurodegeneration [181]. Thus, specific caution should be exercised when selecting methods that use anti-inflammatory approaches for treating HD and other neurodegenerative disorders.

Investigating the measurement and evaluation of central and peripheral immune activation in the progression, diagnosis, and prognosis of HD is a critical area that requires further exploration. Previous reports showed that in HD postmortem tissues, pro-inflammatory microglial and astrocytic cell activation was observed in proportion to the extent of neuronal cell loss [182,183,184]. ^11^C-PBR28 PET-CT imaging can be utilized to evaluate the degree of localized microglia in vivo in HD patients by detecting a translocator protein 18 kDa (TSPO). This technology enables researchers to track microglial activation in living HD patients. Indeed, researchers have found a correlation between the number of CAG repeats in the *HTT* gene and the disease severity, indicating a relationship with the level of microglial activation [185,186,187,188]. Interestingly, studies have shown that microglial activity assessed by PET imaging successfully differentiated HD gene carriers from controls [92,189,190]. Furthermore, it has been reported that LAQ therapy may not influence TSPO expression [148]. Collectively, these imaging investigations indicate the feasibility of examining microglial activation in living patients. The proposition of immune activation in microglia within the CNS can assist in delineating disease progression and assessing the effectiveness of anti-inflammatory interventions. On the other hand, the interaction between the brain’s immune system and the body’s peripheral immune system can lead to peripheral immunological factors influencing brain disorders. Interestingly, pre-manifest HD gene carriers have shown heightened levels of the inflammatory cytokines IL-1β, IL5, IL-6, and IL-10 in their peripheral plasma [92,191,192]. Among them, the level of plasma IL-6 elevated as early as 16 years before HD gene carriers’ symptoms manifested [192]. This peripheral immune activation not only coexists with, but also interacts with, the CNS immunological inflammation. Elevated plasma levels of IL-1β, IL-6, IL-8, and TNF-α were associated with increased microglial activation in the somatosensory cortex [92]. Another study indicated a significant correlation between CSF YKL-40 levels and both UHDRS-TMS, UHDRS-TFC, and the stage of HD [193]. An additional study has reported that as the disease progresses, individuals with HD, who initially have lower levels of TGF-β in their blood, may see a return to normal levels [194]. This indicates that the changes in peripheral inflammatory factors are likely to be dynamic as HD advances. Consequently, monitoring the dynamic changes of these inflammatory factors may have auxiliary significance in the diagnosis and assessment of the disease. To conclude, these findings suggest that neuroinflammation driven by microglia and astrocytes persists throughout the course of HD and may even begin before symptoms appear. However, due to assay limitations and the absence of standardized quantitative criteria for inflammatory markers in HD, these changes are not currently the primary assessment of HD progression. However, preliminary results suggest the emergence of a promising system for this assessment.

Although certain inflammatory targets have shown protective effects in different HD models [195], species’ differences and constraints in animal models make it challenging to replicate human disease pathology and immune responses accurately (Table 2). Dosage effects are another critical factor to consider; for example, different doses of inflammatory proteins may exhibit contrasting protective or harmful effects, and variations in complement and immune response levels among patients may influence treatment efficacy. Moreover, the distinctiveness of each patient’s intrinsic immune response may have a considerable impact on the treatment’s effectiveness. Clinical trials revealed that HD patients with elevated baseline complement component 4a/complement component 4 (C4a/C4) levels showed improvements in cUHDRS and TFC, particularly significant compared to those with lower baseline C4a/C4 at specific time points [140]. Based on these findings, C1q inhibition may be especially beneficial for HD patients with elevated complement activation. With more research conducted to validate the harmful or beneficial roles of inflammatory proteins in HD, hopefully, in the near future, more definitive therapies capable of delaying or halting the progressive neurodegenerative pathology of HD will be available.

## Figures and Tables

**Figure 1 ijms-25-11787-f001:**
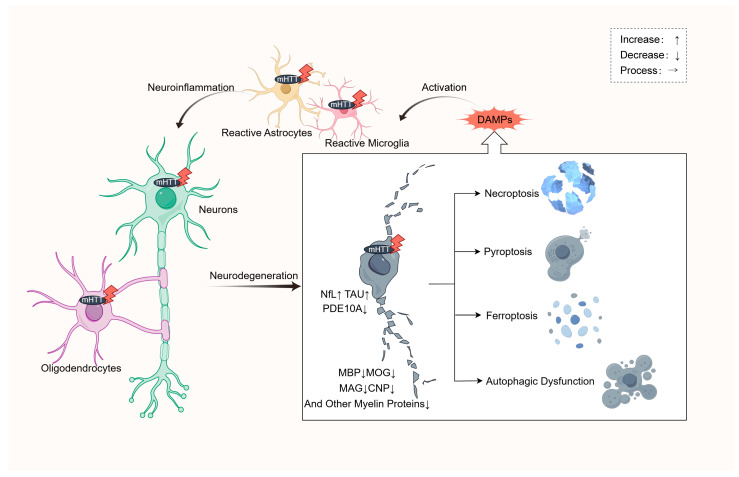
The four different pathological neuronal death mechanisms in HD. Under the influence of activated astrocytes, microglia, and oligodendrocytes with mHTT accumulation, along with intrinsic neuronal mHTT, neuronal structure and function become progressively impaired, resulting in degenerative changes marked by a significant increase in NfL and tau levels, as well as a reduction in PDE10A. The degradation and fragmentation of myelin decrease the expression levels of MBP, MOG, MAG, CNP, and other myelin proteins. As HD progresses, this ultimately leads to neuronal necroptosis, apoptosis, ferroptosis, and autophagic dysfunction, further generating more DAMPs that activate glial cells and amplify inflammation. NfL, neurofilament light chain; PDE10A, phosphodiesterase 10A; DAMPs, damage-associated molecular patterns; MBP, myelin basic protein; MOG, myelin oligodendrocyte glycoprotein; MAG, myelin-associated glycoprotein; CNP, 2′,3′-cyclic nucleotide 3′-phosphodiesterase.

**Figure 2 ijms-25-11787-f002:**
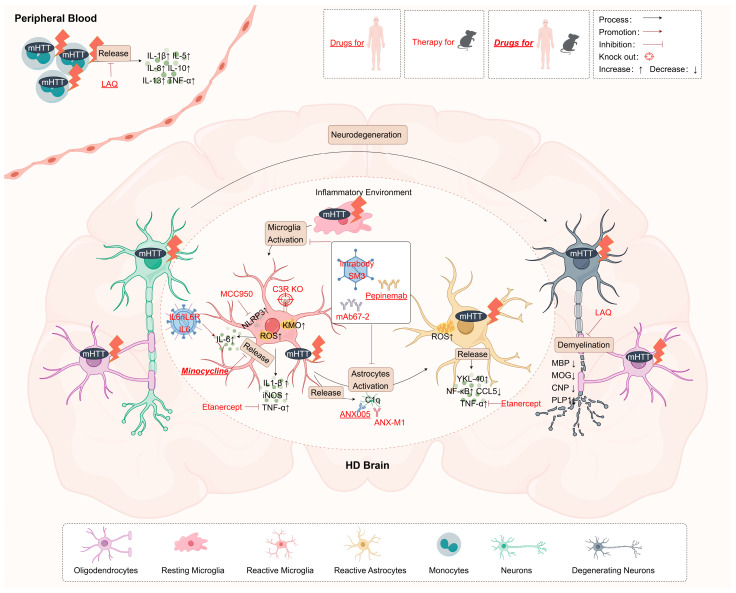
Targeting inflammation in glia and monocytes as drug treatments or gene therapy of HD. Targets highlighted in red indicate targets validated in rodents. Targets highlighted in red and underlined have been validated in patients, while those highlighted in italicized bold red and underlined have been validated in both rodents and patients. ROS, reactive oxygen species; KMO, kynurenine 3-monooxygenase; LAQ, Laquinimod; YKL-40, namely Chitinase 3-like 1; NLRP3, NOD-, LRR-, and pyrin domain-containing protein 3.

**Table 1 ijms-25-11787-t001:** Summary of the mechanisms and effects of immunotherapy or gene therapies in blocking inflammation in HD.

Therapy	Subjects	Routes of Administration	Mechanisms of Action	Results	References
C1q-antibody	ANX-M1 for zQ175 mice	Intraperitoneal (IP) injections	Blocking C1q activity and the classical complement pathway; murine monoclonal antibody for C1q	Improving synaptic function and excitatory signaling input in the striatum	[139]
	ANX005 for HD patients	Intravenous (IV) infusion	Humanized monoclonal antibody for C1q	Safe and well tolerated. There was an improvement in the Composite Unified Huntington’s Disease Rating Scale (cUHDRS) and Total Functional Capacity (TFC), but it did not reach significance	[140]
Semaphorin 4D (SEMA4D) antibody	mAb67-2 for YAC128 mice	IP injections	Inhibiting the SEMA4D pathway to protect the blood–brain barrier and reduce neuroinflammation; murine monoclonal antibody for SEMA4D	Alleviating striatal and cortical atrophy, along with behavioral symptoms, but without affecting motor symptoms	[141]
	Pepinemab (VX15/2503) for HD patients	IV infusion	Humanized monoclonal antibody for SEMA4D	Well tolerated; alleviating caudate nucleus atrophy; motor function did not exhibit significant recovery	[142]
Laquinimod (LAQ)	Peripheral blood monocytes from HD patients	Cell treatments	Reducing peripheral inflammatory responses	Decreasing IL-1β, IL-5, IL-8, IL-10, IL-13, and TNF-α in response to LPS stimulation in manifest HD monocytes; decreasing IL-8, IL-10, and IL-13 in pre-manifest HD	[143]
	YAC128 and R6/2 mice	Oral gavage	Modulating the immune system	Improvement in motor function	[144,145]
	PLP-150Q and YAC128 mice	Oral gavage	Ameliorates myelination deficits in both transcriptional and ultrastructural aspects	Relieving myelin injury	[84,146]
	HD patients	Oral	Modulating the immune system	Mitigating caudate nucleus atrophy, but there was no improvement in motor scores on the Unified Huntington’s Disease Rating Scale–Total Motor Score (UHDRS-TMS) or microglial activation	[147,148]
Minocycline	R6/2 mice	Oral	Reducing pro-inflammatory cytokines and iNOS produced by microglia, thereby mitigating neuroinflammation	Ineffective	[149]
	Quinolinic acid (QA)-induced HD rats	Oral	Consistent with the above	Exerting anti-inflammatory, antioxidant, and neuroprotective abilities	[150]
	HD patients	Oral	Consistent with the above	Safe and well tolerated; unsuccessful in improving motor function	[151,152]
MCC950	R6/2 mice	Oral	NOD-, LRR-, and pyrin domain-containing protein 3 (NLRP3) inhibitor; inhibiting the NLRP3 inflammasome pathway in microglia	Reducing IL-1β and IL-18; rescuing motor dysfunction and neuronal survival	[153]
IL-6 or IL-6/IL-6R (IL-6 receptor)-overexpressing lentivirus	QA-induced HD rats	Stereotactic intrastriatal injection	Increasing IL-6 to nurture neuron	Neuroprotective effects	[154]
Etanercept	R6/2 mice	IP injections or IV infusion	TNF-α inhibitor	Rescuing brain atrophy; did not improve related motor functions and cognitive deficits.	[155]
Tominersen	HD patients	Intrathecal injection	Targeting the *HTT* gene’s mRNA to decrease mHTT	The Phase 3 study of the GENERATION HD1 trial was halted for its disappointing performance; the GENERATION HD2 trial is ongoing; the potential risk of CNS inflammation	[156,157,158]
AMT-130	HD patients	Stereotaxic intrastriatal injection	Targeting the *HTT* gene’s mRNA to decrease mHTT	Reduced mHTT protein levels; improved motor function; and decreased levels of CSF NfL; The results of its combination with immunosuppression are still unknown	[159]
CRISPR/CAS9	HD knock-in (KI) pig	Stereotaxic intrastriatal injection	Mutant *HTT* gene ablation	The depletion of mHTT; decreasing *IL1B* and *NFKB* mRNA in the striatum.	[160]
Adeno-associated virus (AAV)-packaged SM3 intrabody	CAG140 KI mice	Stereotaxic intrastriatal injection or IV infusion	Target intracellular mHTT for lysosomal degradation	Reduction of mHTT protein; decreasing reactive microglia and astrocytes in the striatum	[161]

**Table 2 ijms-25-11787-t002:** Limitations and species heterogeneity of HD animal models.

HD Animal Models	Limitations	References
Transgenic HD rats	Phenotypes develop slowly	[196]
QA-induced HD rats	Chemically induced neuronal damage; no genetic component	[197]
Almost all HD mice models	Unable to replicate dystonia and chorea; no apparent neuronal death	[17,196]
R6/2 mice	Mice may die before significant glial inflammation develops	[198,199,200]
YAC128 mice	The hippocampus lacked neuronal intranuclear inclusions (NIIs)	[201]
BACHD mice	More than 90% of the inclusions were extranuclear; inclusions are more prevalent in the cortex rather than the striatum	[17,202]
zQ175 mice	No neuronal cell death	[203]
HD KI mice (CAG140, HdhQ72-80, HdhQ111 mice)	Mild phenotypes and no neuronal cell death	[204,205,206]
Large animals	Quadrupeds have unique gate and balancing mechanisms that differ from humans	[207]
Transgenic sheep	Absence of a strong or obvious behavioral phenotype	[208]
Transgenic monkeys	Generally single-birth and long reproductive cycles increase the cost and difficulty of establishing a study cohort	[209,210]
Transgenic pigs	Limited lifespans due to the intensity of their clinical symptoms	[211]

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
