# Peer review of "Neuroinflammatory Proteins in Huntington’s Disease: Insights into Mechanisms, Diagnosis, and Therapeutic Implications"

_ijms, 2024, doi:10.3390/ijms252111787_

Round 1
Reviewer 1 Report
Comments and Suggestions for Authors
Li et al. reported on the glial cell and peripheral immune responses triggered by mHTT in HD, and identified several neuronal damage biomarkers detectable following inflammatory stimuli. This is an insightful review article that thoroughly discusses the mechanisms, diagnosis, and treatment of HD, using neuroinflammation as a central theme. The paper is well-written, but there are a few comments that could help further improve the manuscript.
1. The comparison of Immunotherapy in blocking inflammation shown in Table 1 with antisense oligonucleotides and RNAi, which are already in clinical application, should be described in more detail. Then, the advantages and/or synergistic effects using both immunotherapy and nucleic acid therapy should be mentioned.
2. For each immunotherapy listed in Table 1, the route of administration should also be indicated.
Author Response
Dear Editor and Reviewer,
We are very grateful for the opportunity to revise our manuscript. We like to thank the editor and reviewers for taking the time and efforts to provide insightful comments and suggestions. We have carefully read the comments and suggestions, and revised the paper accordingly.
Below are the point-by-point responses to the reviewers’ comments. All changes to the manuscript are highlighted in yellow.
Reviewer 1
Li et al. reported on the glial cell and peripheral immune responses triggered by mHTT in HD, and identified several neuronal damage biomarkers detectable following inflammatory stimuli. This is an insightful review article that thoroughly discusses the mechanisms, diagnosis, and treatment of HD, using neuroinflammation as a central theme. The paper is well-written, but there are a few comments that could help further improve the manuscript.
We thank the reviewer’s positive comments and suggestions, we have modified the content as listed below.
- The comparison of Immunotherapy in blocking inflammation shown in Table 1 with antisense oligonucleotides and RNAi, which are already in clinical application, should be described in more detail. Then, the advantages and/or synergistic effects using both immunotherapy and nucleic acid therapy should be mentioned.
Response: Thanks for the valuable comments. We have carefully addressed your suggestions by updating Table 1 to include additional details on the two therapies. Furthermore, we have expanded our discussion to clarify the potential risks of CNS inflammation associated with Tominersen (an antisense oligo), and we have provided an overview of the ongoing trials involving AMT-130 and immunosuppressive agents in the text lines 612-646, referencing [156-158, 174-177]. As the outcomes of the combination of immunotherapy and AMT-130 are still pending, which we cannot comment the result.
- For each immunotherapy listed in Table 1, the route of administration should also be indicated.
Response: This is a good suggestion. We have added the routes of administration in the table 1.
Reviewer 2 Report
Comments and Suggestions for Authors
This is an interesting and well-structured manuscript, although largely descriptive, which has been competently performed by authors with a good record of previous publications in this topic. I am including a few suggestions to lend further support to the major conclusions achieved in this review. This review is focused on a qualitative assessment of the results obtained using cellular and genomic approaches with Huntington’s disease (HD) model animals and patient tissues, and highlights the limitations of the studies with animal models to replication of human disease pathology and immune responses. Due to the relevance of the latter point for diagnosis and therapeutic treatments of HD, this manuscript can be improved including a Table summarizing the most notable species differences and limitations in the section 6 -Future Perspectives. In addition, a more in-dept critical analysis of the scientifically relevant points indicated next could be very helpful to make this review less descriptive. The quantitative assessment of the level of the various glial cell and peripheral immune responses triggered by mHTT is a missing point in this review, despite that it is particularly relevant for the defense/neurotoxicity balance in the brain. Therefore, the activation threshold levels of the various glial cell in HD-affected regions of the brain and peripheral immune responses that modify the progression and prognosis of HD, and/or extend survival are particularly relevant for therapeutic treatments of HD. Are these threshold levels already established? Can the authors comment on these points where appropriate in the text and in the section 6-Future Perspectives? Other proposed corrections (minor corrections) are included below.
Minor corrections list:
1. Please, revise the following statement of lines 112-113: “microglia derived from HD patient’s pluripotent stem cells produce reactive oxygen species (ROS), leading to oxidative stress (OS) damage and enhanced apoptosis.” Should not be necroptosis or apoptosis and necroptosis, instead of solely apoptosis for coherence with the content of the section 4. Biological Markers Associated with Inflammatory Damage in HD?
2. The following sentence of lines 164-168 is too long and somewhat ambiguous, please, shorten it and modify for the sake of clarity: “Intriguingly, in this model, stereotaxic injection of adeno-associated virus (AAV)-mediated overexpression of heat shock protein 70 -binding protein 1 (HspBP1) in astrocytes, leading to increased mHTT accumulation in astrocytes, exacerbated neurological symptoms but did not trigger neurodegeneration, suggesting the pivotal but indirect contributors of non-cell-autonomous toxicity”.
3. The following paragraph of lines 362-372 is controversial, as the sentences of the final part of the paragraph (lines 378-372) look somewhat contradictory with the previous sentences. For the sake of clarity, please, modify this paragraph: “In recent investigations, researchers have explored whether KP metabolites associated with KMO can serve as diagnostic biomarkers for HD through analyses of biofluids [108]. Although no significant differences in KP metabolites were observed in the cerebrospinal fluid (CSF) and blood of pre-symptomatic and symptomatic HD patients, compared to healthy controls, suggesting that KP metabolites may not serve as reliable biomarkers for differentiating between the stages of the disease [108]. However, the researchers concluded that this does not negate the therapeutic potential of KMO [108]. This finding expands upon previous studies by confirming that KP metabolites can be effectively quantified in biofluids outside the brain parenchyma, providing insights for monitoring disease progression and assessing the biological efficacy of potential therapy for HD [108].”
4. Please, revise the following sentence of the lines 161-162: “astrocytes expression of mHTT resulting in excitotoxicity and reduced synaptic plasticity”. Do you mean “results in” instead of “resulting in”.
5. Please, revise the following sentence of the lines 177-178: “indicating that the extent of the mutation governing the astrocyte behavior”. Do you mean: “governs” instead of “governing”?
6. Please, revise the following sentence of the lines 492-493: “which may result 492 from different physiological, metabolic conditions.” Do you mean: “physiological and/or metabolic conditions”?
7. Please, correct the typo errors indicated below:
-“Gila-“ should read “Glia-“ (Heading of section 2, line 77)
-Line 393: “Phosphodiesterase” instead of “phosphodiesterase”.
-Line 553: “researchers (or investigators)” instead of “researcher”
Author Response
This is an interesting and well-structured manuscript, although largely descriptive, which has been competently performed by authors with a good record of previous publications in this topic. I am including a few suggestions to lend further support to the major conclusions achieved in this review. This review is focused on a qualitative assessment of the results obtained using cellular and genomic approaches with Huntington’s disease (HD) model animals and patient tissues, and highlights the limitations of the studies with animal models to replication of human disease pathology and immune responses. Due to the relevance of the latter point for diagnosis and therapeutic treatments of HD, this manuscript can be improved including a Table summarizing the most notable species differences and limitations in the section 6 -Future Perspectives. In addition, a more in-dept critical analysis of the scientifically relevant points indicated next could be very helpful to make this review less descriptive. The quantitative assessment of the level of the various glial cell and peripheral immune responses triggered by mHTT is a missing point in this review, despite that it is particularly relevant for the defense/neurotoxicity balance in the brain. Therefore, the activation threshold levels of the various glial cell in HD-affected regions of the brain and peripheral immune responses that modify the progression and prognosis of HD, and/or extend survival are particularly relevant for therapeutic treatments of HD. Are these threshold levels already established? Can the authors comment on these points where appropriate in the text and in the section 6-Future Perspectives? Other proposed corrections (minor corrections) are included below.
Response: We thank the reviewer’s insightful comments and have revised the manuscript according to the comments and suggestions. We added Table 2 to summarize the most significant species differences and limitations of animal models of HD.
We also discussed the close relationship between central and peripheral immune activation in the progression, diagnosis, and prognosis of HD in lines 690-726. Many studies have attempted to demonstrate how to quantify TSPO in the brain or detect inflammatory factors in peripheral biofluid to determine the extent of inflammation in the HD brain, and they have found that microglial activation is closely related to the extent of neuronal death [ref 182-194]. However, due to the limitations of the assays and the lack of harmonized quantitative criteria for inflammatory indicators in HD, their changes are not currently the main assessment of the HD process, but the preliminary results hint at the emergence of a promising system for this assessment.
In addition, to make this review less descriptive, we have added some more comments and discussions throughout the paper
Minor corrections list:
- Please, revise the following statement of lines 112-113: “microglia derived from HD patient’s pluripotent stem cells produce reactive oxygen species (ROS), leading to oxidative stress (OS) damage and enhanced apoptosis.” Should not be necroptosis or apoptosis and necroptosis, instead of solely apoptosis for coherence with the content of the section 4. Biological Markers Associated with Inflammatory Damage in HD?
Response:In accordance with the protocol provided by the Promega’s Real Time-Glo Annexin V Apoptosis Assay kit, as described in the methods section of the ref [36], it is noted that this kit lacks DNA dyes capable of penetrating cells. Consequently, it is designed to detect early apoptosis rather than secondary necrosis or necroptosis. This limitation implies that the specific mode of cell death—whether necroptosis or just apoptosis—could not be definitively determined in the study. The research focuses on the death of microglia, and the distinction between necroptosis and conventional apoptosis remains unclear. Our subsequent discussion pertains to the mode of death following neuronal degeneration, which may not be in direct conflict with the findings presented. To ensure precision and maintain contextual consistency, we have engaged in a thorough examination of the topic and have included this discussion in lines 159-187. We hope this addition provides clarity and enhances the overall understanding of the study.
(The article and instructions are as follows or in the Word file)
O'Regan, G. C.; Farag, S. H.; Casey, C. S.; Wood-Kaczmar, A.; Pocock, J. M.; Tabrizi, S. J.; Andre, R. Human Huntington's disease pluripotent stem cell-derived microglia develop normally but are abnormally hyper-reactive and release elevated levels of reactive oxygen species. J Neuroinflammation 2021, 18, (1), 94.
- The following sentence of lines 164-168 is too long and somewhat ambiguous, please, shorten it and modify for the sake of clarity: “Intriguingly, in this model, stereotaxic injection of adeno-associated virus (AAV)-mediated overexpression of heat shock protein 70 -binding protein 1 (HspBP1) in astrocytes, leading to increased mHTT accumulation in astrocytes, exacerbated neurological symptoms but did not trigger neurodegeneration, suggesting the pivotal but indirect contributors of non-cell-autonomous toxicity”.
Response:Thanks for the kind suggestion. We have modified long sentences to make them more fluent and understandable in lines 242-247.
- The following paragraph of lines 362-372 is controversial, as the sentences of the final part of the paragraph (lines 378-372) look somewhat contradictory with the previous sentences. For the sake of clarity, please, modify this paragraph: “In recent investigations, researchers have explored whether KP metabolites associated with KMO can serve as diagnostic biomarkers for HD through analyses of biofluids [108]. Although no significant differences in KP metabolites were observed in the cerebrospinal fluid (CSF) and blood of pre-symptomatic and symptomatic HD patients, compared to healthy controls, suggesting that KP metabolites may not serve as reliable biomarkers for differentiating between the stages of the disease [108]. However, the researchers concluded that this does not negate the therapeutic potential of KMO [108]. This finding expands upon previous studies by confirming that KP metabolites can be effectively quantified in biofluids outside the brain parenchyma, providing insights for monitoring disease progression and assessing the biological efficacy of potential therapy for HD [108].”
Response:Thanks for your good suggestion, and we have added a relevant reference and discussed the possible reasons for the different results of the KP metabolites’ studies in HD blood, presented in lines 443-454.
- Please, revise the following sentence of the lines 161-162: “astrocytes expression of mHTT resulting in excitotoxicity and reduced synaptic plasticity”. Do you mean “results in” instead of “resulting in”.
Response:Thanks. We've revised the words into “results in” in line 235.
- Please, revise the following sentence of the lines 177-178: “indicating that the extent of the mutation governing the astrocyte behavior”. Do you mean: “governs” instead of “governing”?
Response:Thanks. We've revised the word into “governs” in line 256.
- Please, revise the following sentence of the lines 492-493: “which may result 492 from different physiological, metabolic conditions.” Do you mean: “physiological and/or metabolic conditions”?
Response:Thanks. We've changed the words into “physiological and metabolic conditions” in line 587.
- Please, correct the typo errors indicated below:
-“Gila-“ should read “Glia-“ (Heading of section 2, line 77). It’s done. Line 124
-Line 393: “Phosphodiesterase” instead of “phosphodiesterase”. It’s done. Line 475
-Line 553: “researchers (or investigators)” instead of “researcher”. It’s done, line 678
Reviewer 3 Report
Comments and Suggestions for Authors
In the study “Neuroinflammatory Proteins in Huntington’s Disease: Insights into Mechanisms, Diagnosis, and Therapeutic Implications” by Li et al., the authors attempted to describe the role of inflammation and its therapeutic significance in the development and treatment of Huntington’s disease.
The study is thorough, detailed, and interesting. Thus, it merits publication.
There are few suggestions that can clarify some issues raised in the paper.
Introduction:
The authors explained the pathology of HD in both preclinical animal models and in the HD patients. In the last paragraph the authors state that the role of inflammation has already been widely reported and summarized and clarify that this review focuses on the glial responses and peripheral immune reactions triggered by mHTT in preclinical experiments or clinical trials. However, brief paragraph summarizing the current knowledge of neuroinflammation in HD will be beneficial to the reader and will facilitate further reading of the article.
Line 47: The authors state that the initial stage of HD is critical for the potential therapeutic strategies. They also define the three stages of the disease progression. It will be helpful to give, if possible, the estimated duration of each of these stages. For example, the presymptomatic phase in Alzheimer’s disease can last several decades. Is that the case in the HD or this presymptomatic phase has different duration? Are there differences in the length of this phase in regards to the number of CAG repeats?
Line 309: in the chapter 4. Biological Markers Associated with Inflammatory Damage in HD, the authors introduce Figure 1 in the first paragraph. However, Nfl, tau, and PDE10A markers depicted in the Figure 1 and mentioned in the legend, are explained much later in the text creating confusion. A different positioning of the Figure 1 in the text can remedy that.
Line 310: The sentence - In NNDs, pathological neuronal death mechanisms have been confirmed to include necroptosis, pyroptosis, ferroptosis, and autophagy – imply that autophagy is pathological. It should be written here (as it is written in the subsequent text) that it is in fact autophagy dysfunction.
Author Response
Dear Editor and Reviewer,
We are very grateful for the opportunity to revise our manuscript. We like to thank the editor and reviewers for taking the time and efforts to provide insightful comments and suggestions. We have carefully read the comments and suggestions, and revised the paper accordingly.
Below are the point-by-point responses to the reviewers’ comments. All changes to the manuscript are highlighted in yellow.
In the study “Neuroinflammatory Proteins in Huntington’s Disease: Insights into Mechanisms, Diagnosis, and Therapeutic Implications” by Li et al., the authors attempted to describe the role of inflammation and its therapeutic significance in the development and treatment of Huntington’s disease.
The study is thorough, detailed, and interesting. Thus, it merits publication.
There are few suggestions that can clarify some issues raised in the paper.
Introduction:
The authors explained the pathology of HD in both preclinical animal models and in the HD patients. In the last paragraph the authors state that the role of inflammation has already been widely reported and summarized and clarify that this review focuses on the glial responses and peripheral immune reactions triggered by mHTT in preclinical experiments or clinical trials. However, brief paragraph summarizing the current knowledge of neuroinflammation in HD will be beneficial to the reader and will facilitate further reading of the article.
Response: Thank you for your insightful suggestions. We've added the “The Summary and Highlights” in the introductory section.
Line 47: The authors state that the initial stage of HD is critical for the potential therapeutic strategies. They also define the three stages of the disease progression. It will be helpful to give, if possible, the estimated duration of each of these stages. For example, the presymptomatic phase in Alzheimer’s disease can last several decades. Is that the case in the HD or this presymptomatic phase has different duration? Are there differences in the length of this phase in regards to the number of CAG repeats?
Response: This is a very good suggestion. We have added the related description of different stages in line 45-70. We also provided a more detailed description of the lasting time of different stages in HD and it does depend on the number of CAG repeats.
Line 309: in the chapter 4. Biological Markers Associated with Inflammatory Damage in HD, the authors introduce Figure 1 in the first paragraph. However, Nfl, tau, and PDE10A markers depicted in the Figure 1 and mentioned in the legend, are explained much later in the text creating confusion. A different positioning of the Figure 1 in the text can remedy that.
Response: Thanks for the good suggestion. We have moved Figure 1 to page 16.
Line 310: The sentence - In NNDs, pathological neuronal death mechanisms have been confirmed to include necroptosis, pyroptosis, ferroptosis, and autophagy – imply that autophagy is pathological. It should be written here (as it is written in the subsequent text) that it is in fact autophagy dysfunction.
Response: We have revised this to autophagy dysfunction line 391